# KDEL Receptor Trafficking to the Plasma Membrane Is Regulated by ACBD3 and Rab4A-GTP

**DOI:** 10.3390/cells12071079

**Published:** 2023-04-04

**Authors:** Chuanting Tan, Yulei Du, Lianhui Zhu, Shuaiyang Jing, Jingkai Gao, Yi Qian, Xihua Yue, Intaek Lee

**Affiliations:** 1School of Life Science and Technology, ShanghaiTech University, Shanghai 201210, China; 2University of Chinese Academy of Sciences, Beijing 101408, China

**Keywords:** KDEL receptor, ACBD3, Rab4A, Rab11A, Golgi, plasma membrane

## Abstract

KDEL receptor-1 maintains homeostasis in the early secretory pathway by capturing and retrieving ER chaperones to the ER during heavy secretory activity. Unexpectedly, a fraction of the receptor is also known to reside in the plasma membrane (PM), although it is largely unknown exactly how the KDEL receptor gets exported from the Golgi and travels to the PM. We have previously shown that a Golgi scaffolding protein (ACBD3) facilitates KDEL receptor localization at the Golgi via the regulating cargo wave-induced cAMP/PKA-dependent signaling pathway. Upon endocytosis, surface-expressed KDEL receptor undergoes highly complex itineraries through the Golgi and the endo-lysosomal compartments, where the endocytosed receptor utilizes Rab14A- and Rab11A-positive recycling endosomes and clathrin-decorated tubulovesicular carriers. In this study, we sought to investigate the mechanism through which the KDEL receptor gets exported from the Golgi en route to the PM. We report here that ACBD3 depletion results in greatly increased trafficking of KDEL receptor to the PM via Rab4A-positive tubular carriers emanating from the Golgi. Expression of constitutively activated Rab4A mutant (Q72L) increases the surface expression of KDEL receptor up to 2~3-fold, whereas Rab4A knockdown or the expression of GDP-locked Rab4A mutant (S27N) inhibits KDEL receptor targeting of the PM. Importantly, KDELR trafficking from the Golgi to the PM is independent of PKA- and Src kinase-mediated mechanisms. Taken together, these results reveal that ACBD3 and Rab4A play a key role in regulating KDEL receptor trafficking to the cell surface.

## 1. Introduction

The Golgi apparatus plays essential roles in receiving newly synthesized proteins from the endoplasmic reticulum (ER) and delivering them to targeted destinations [1,2]. During this highly coordinated process, some ER resident proteins are transported to the Golgi with nascent proteins and subsequently retrieved by the ER. A seven-transmembrane Golgi protein, KDEL receptor (KDELR) recognizes a ‘KDEL’ signal motif, present in the C-termini of most soluble ER proteins and transports them from the Golgi to ER via coat protein complex I (COPI) vesicles [3,4,5].

At steady state, KDELR mostly localizes to the Golgi via a scaffold protein, acyl-CoA binding containing protein 3 (ACBD3), and maintains dynamic homeostasis in the early secretory pathway [6,7]. Previous studies have reported that a small fraction of KDELR resides on the cell surface as well and circulates between the plasma membrane and the Golgi via a complex trafficking itinerary [8,9,10,11]. Similar to canonical cell surface receptors, endocytosed KDELR is recycled to the plasma membrane via Rab GTPase-mediated early and recycling endosomal pathways. Rab proteins are critical components of the machinery for membrane trafficking and are involved in all steps from the initial cargo sorting to the final stage of fusion with destination membranes. Rab GTPases cycle between GTP-bound active and GDP-bound inactive states, which are catalyzed by guanine nucleotide exchange factors (GEFs) and GTPase-activating proteins (GAPs) [12,13,14]. Interestingly, studies have shown that the transport between the *trans*-Golgi network (TGN) and endosomes are bidirectional, and transport in each direction is regulated by Rab GTPases [15,16,17,18]. It remains elusive whether KDELR exits the Golgi and transports to the plasma membrane via a Rab protein-mediated endosomal route.

The Golgi-to-ER retrograde transport of KDEL-containing ER proteins requires protein kinase A (PKA) phosphorylation of the C-terminus of KDELR, which in turn allows the recruitment of ArfGAP1/COPI proteins to the Golgi membranes and ensures the retrieval of KDELR ligands [19,20]. Our previous study showed that ACBD3, a putative PKA anchoring protein (AKAP), regulates KDELR-PKA interactions and the ER traffic-induced retrograde transport of KDELR via ADP-ribosylation factor 1 (Arf1)-dependent tubulovesicular carriers [6]. The Src tyrosine kinase has been shown to regulate KDELR retrograde transport to the ER [21]. It has been reported that the chaperone–KDELR interaction stimulates a Golgi pool of Src-family kinases, which then, in turn, activate anterograde traffic through the Golgi [22]. By now, there are several pathways involved in regulating KDELR retrograde trafficking, including the KDEL ligand, Src, PKA, ARFGAP1 and ACBD3. However, it is unknown whether these mechanisms are involved in regulating KDELR transportation to the plasma membrane.

Here, we studied the role of the KDEL ligand, Src, PKA, ARFGAP1 and ACBD3 in regulating the cell surface expression of KDELR. We show that only ACBD3 depletion greatly increases the cell surface expression level of KDELR. Live cell imaging results indicate that the Golgi export of KDELR was carried out mostly by Rab4A-labelled tubules. The overexpression of constitutively active Rab4A (Q72L) accelerated KDELR trafficking to the plasma membrane, whereas GDP-locked Rab4A mutant (S27N) inhibited KDELR expression at the PM. Furthermore, Rab4A depletion inhibits Golgi-derived transport carrier formation in ACBD3-depleted cells. These results provide a novel insight into the mechanistic underpinning of the Golgi-to-cell surface trafficking of KDELR.

## 2. Materials and Methods

### 2.1. Cell Culture and Transfection

HeLa and HeLa S3 cells obtained from ATCC were grown in Dulbecco’s modified Eagle medium (DMEM, Meilunbio, Dalian, China) supplemented with 10% fetal bovine serum (FBS, EXCELL). HepG2, DU145 and HT-1080 (Stem Cell Bank, Chinese Academy of Sciences) were grown in Minimum Essential Medium (MEM, Meilunbio) supplemented with 10% FBS. A549 (Stem Cell Bank, Chinese Academy of Sciences) was grown in F-12K (Meilunbio) supplemented with 10% FBS. U-2 OS (Meilunbio) was grown in McCoy’s 5A (Meilunbio) supplemented with 10% FBS. HeLa S3 cells were authenticated via STR profiling. The authentication of other cell lines are provided by the provider. All cell lines were routinely tested for mycoplasma contamination and were negative. The transfection of DNA constructs and siRNAs was performed using Lipofectamine 2000 and RNAi-MAX (ThermoFisher, Carlsbad, CA, USA), respectively, according to the manufacturer’s instructions. For DNA expression, cells were transfected 18 h before immunofluorescence (IF) experiments. For siRNA knockdown, cells were transfected 72 h before experiments.

### 2.2. Antibodies, Reagents, siRNAs, shRNA and CRISPR Knockout

The following antibodies were used in this study: anti-ACBD3 (HPA015594, Sigma-Aldrich, St. Louis, MO, USA), anti-mCherry (ab167453, Abcam, Cambridge, MA, USA), anti-GFP (ab6556, Abcam, MA, USA), anti-Flag (F1804, Sigma-Aldrich, MO, USA), anti-Myc-tag (2278S, CST, MA, USA), anti-GFP (ab6556, Abcam, MA, USA), anti-ARFGAP1 (ab204405, Abcam, MA, USA), anti-PKA-cα (#4782, Cell Signaling Technology, Danfoss, MA, USA), anti-EGFR (4267s, Cell Signaling Technology, MA, USA) and anti-GAPDH-HRP (HRP-60004, Proteintech, Chicago, IL, USA).

The siRNAs were custom designed by Shanghai GenePharma, China. The sequence of the non-targeting control siRNA was UUCUCCGAACGUGUCACGU. The other sequences were as follows: ACBD3 siRNA-1: GCAUUAGAGUCACAGUUUA; ACBD3 siRNA-2: GCUGAAGUUACAUGAGCUA; ARFGAP1 siRNA-1: AAGGUGGUCGCUCUGGCCGAAG; ARFGAP1 siRNA-2: GCAACAUAGACCAGAGCUU; and Rab4A siRNA: GGUUAACAGAUGCCCGAAU. We combined siRNA-1 and siRNA-2 to get a high knockdown efficiency for the experiments. The stable knockdown of ACBD3 was achieved by infecting target cells using a lentivirus expression of ACBD3-shRNA (GCTGAAGTTACATGAGCTACA). The lentivirus was packaged and commercially provided by Shanghai GenePharma, China. Cells were infected with the lentivirus expressing ACBD3-shRNA using Polybrene (Sigma) overnight. Two days after infection, the cells were cultured in puromycin (0.3–1 μg/mL, ThermoFisher) for 2 weeks.

CRISPR knockout of ACBD3 in HT1080, HeLa and HeLa S3 cells was performed as described previously [6].

### 2.3. Cell Surface Biotinylation

Cells grown in 6-well plates to 80% confluency were transfected using 0.8 μg of plasmid DNA and 3 μL of Lipofectamine 2000 for 18 h. During the biotinylation procedure, all reagents and cell cultures were kept on ice. Cells were washed twice in ice-cold PBS and subsequently incubated in 1 mL/well of a 1 mM Sulfo-NHS-LC-Biotin (APExBIO) in PBS solution for 30 min on ice. The cells were then washed in quenching buffer (100 mM glycine in PBS), and incubated in 1 ml/well of quenching buffer for 15 min on ice. The cells were washed twice with PBS and then lysed in 300 μL of RIPA lysis buffer (50 mM Tris, pH 7.4, 150 mM NaCl, 0.1% SDS, 1% (*v*/*v*) Triton X-100, 0.5% (*w*/*v*) deoxycholate, and protease inhibitor cocktail (Roche, IN, USA)). Lysates were incubated for 20 min on ice and sonicated for 20 s. Finally, the lysate was centrifuged at 4 °C for 10 min at 15,000× *g*. Supernatants were incubated with 40 μL of Streptavidin Agarose beads (S1638, Sigma Aldrich) with constant rocking for 1 h at 4 °C. The samples were washed three times with PBS, then eluted with 2x SDS-sample buffer for 10 min at 95 °C and used for Western blot.

### 2.4. Immunoblotting

For immunoblotting, proteins were separated using SDS-PAGE (Genscript) and transferred onto nitrocellulose membranes (Amersham). Membranes were probed with specific primary antibodies and then with peroxidase-conjugated secondary antibodies (Jackson ImmunoResearch, West Grove, PA, USA). The bands were visualized with chemiluminescence (Clarity Western ECL Substrate, Bio-Rad, Hercules, CA, USA) and imaged using a ChemiDoc Touch imaging system (Bio-Rad). Representative blots are shown from several experiments.

### 2.5. Immunofluorescence Staining and Confocal Microscopy

Cells grown on glass coverslips in 24-well plates were fixed for 10 min with 4% paraformaldehyde (PFA), permeabilized in permeabilization Buffer (0.3% Igepal CA-630, 0.05% Triton-X 100, 0.1% IgG-free BSA in PBS) for 3 min, then blocked in blocking buffer (0.05% Igepal CA-630, 0.05% Triton-X 100, 5% normal goat serum in PBS) for 60 min. Primary and secondary antibodies were applied in blocking buffer for 1 h. The nucleus was stained with Hoechst-33342 (sc-200908, Santa cruz Biotechnology, Dallas, TX, USA). Cells were washed three times with wash buffer (0.05% Igepal CA-630, 0.05% Triton-X 100, 0.2% IgG-free BSA in PBS) and twice with PBS. Coverslips were mounted using ProLong Gold Antifade Reagent (ThermoFisher).

For the cell surface staining of 3×Flag-KDELR1-mCherry, HeLa S3 cells were seeded on a 24-well glass-bottom plate (Cellvis) coated with fibronectin (Millipore, Burlington, MA, USA). After 18 h transfection with 3×Flag-KDELR1-mCherry and indicated plasmid, cells were incubated with anti-Flag antibody (Sigma) in DMEM with 2% FBS at 4 °C for 1 h. The cells were washed twice with ice-cold PBS, and then fixed using 4% PFA. After washing three times with PBS, the cells were incubated with secondary antibody for 1 h. Cells were washed three times and then were imaged with a 63× objective on a Zeiss LSM 880 confocal microscope.

### 2.6. Live Cell Imaging

For photoactivation experiments, HT1080 WT or ACBD3 stably knockdown cells were seeded on a glass-bottom dish (35 mm diameter, Cellvis) coated with fibronectin (Millipore). After 18 h co-transfection with KDELR1-PA-GFP and indicated mCherry-Rab, cells were imaged with a 63× objective on a Zeiss LSM 880 confocal microscope in an atmosphere of 5% CO_2_ at 37 °C. Photoactivating of KDELR1-PA-GFP in the Golgi was achieved using a 405 nm laser. Images were acquired every 5 s for 5 min.

For live cell imaging, HeLa WT or ACBD3-KO cells were seeded on a glass-bottom dish (35 mm diameter, Cellvis) coated with fibronectin (Millipore). After 18 h co-transfection with GT-GFP and mCherry-Rab4A, cells were imaged with a 63× objective on a Zeiss LSM 880 ariyscan confocal microscope in an atmosphere of 5% CO_2_ at 37 °C. Images were acquired every 5 s for 5 min.

### 2.7. Image Processing and Statistical Analysis

Results are displayed as mean ± SD (standard deviation) of the results from each experiment or dataset, as indicated in figure legends. All statistical tests were performed using Student’s T tests or ANOVA (GraphPad, Prism). Significance values were assigned to specific experiments. N (number of individual experiments) is noted in the figure legends.

## 3. Results

### 3.1. Surface Expression of KDELR Is Greatly Increased in ACBD3-Depleted Mammalian Cell Lines

To study the relationship between KDELR retrograde trafficking and its cell surface localization, we conducted a small scale screening of those mechanisms involved in regulating KDELR retrograde trafficking to check their roles in regulating KDELR transporting to the plasma membrane. First, we tested the surface expression of KDELR1-mCherry upon the ligand-induced re-location of KDELR to the ER. We co-transfected a secretory cargo fused to the C-terminal KDEL sequence (hGH-GFP-KDEL) with KDELR1-mCherry overnight in HT1080 cells, followed by a surface biotinylation protocol to examine the surface expression of KDELR1 in these cells.

The results showed that expression of hGH-GFP-KDEL caused a decreased surface expression of KDELR1-mCherry compared with GFP or hGH-GFP control (Figure 1A). Both Src and PKA have been shown to regulate KDELR retrograde transport to the ER. Thus, we tested surface expression of KDELR1-mCherry on the overexpression of a constitutively active Src(E381G) mutant or PKA-Cα. The results showed that the surface level of KDELR1-mCherry was moderately increased after Src(E381G) overexpression, whereas the surface expression of KDELR1-mCherry was moderately decreased after PKA-Cα overexpression, compared to the control (Figure 1B,C). It has been reported that ARFGAP1 facilitates the sorting of KDELR1 and the formation of COPI vesicles [23,24,25]. To determine the potential role of ARFGAP1 in regulating KDELR surface expression, we tested the surface expression of KDELR1-mCherry after RNAi-mediated ARFGAP1 knockdown for 48 h. The results showed that ARFGAP1 depletion has no effect on the surface expression of KDELR1-mCherry, compared to the control (Figure 1D).

In our earlier study, we reported that ACBD3, a Golgi scaffolding protein, plays a key role in suppressing PKA activity on KDELR, thereby facilitating the Golgi localization of KDELR under a steady-state condition [6]. To study whether ACBD3 may be involved in the receptor export from the Golgi to the PM, we tested the surface expression of KDELR1-mCherry in ACBD3 knockout HT1080 cells using a surface biotinylation protocol [11]. The results showed that ACBD3 knockout greatly increases the surface expression of KDELR1-mCherry (Figure 1E).

To further confirm this result, we tested the surface expression of KDELR1-mCherry after ACBD3 knockdown in several other cell lines. To this end, five human cell lines including HeLa-S3, human lung cancer cell A549, human prostate cancer cell DU145, human hepatoma HepG2 and human osteosarcoma U-2 OS were treated with shRNA lentivirus to stably knock down ACBD3. We then transiently transfected the cells with KDELR1-mCherry overnight, followed by a surface biotinylation protocol to examine the surface expression of KDELR1 in these cells. The results showed that all five cell lines displayed a 1.5~3-fold increase in the surface expression of KDELR1-mCherry upon ACBD3 depletion (Figure 2A–E), while the expression of shRNA-resistant myc-ACBD3 completely restored KDELR1 surface expression to that of the control cells (Figure 2F). To confirm this finding by confocal imaging, we transiently co-transfected ACBD3 KD HeLa-S3 cells as well as ACBD3 KO cells [6] with lumenally 3xFlag-tagged KDELR-mCherry (Figure 2G) and YFP-GL-GPI (as a PM marker) overnight, followed by indirect immunostaining using anti-Flag antibody, in order to measure the surface expression of KDELR1 in these cells. Upon examination with a confocal microscope, both ACBD3 KD and KO resulted in a significantly increased surface expression of KDELR1 (Figure 2H), further confirming the earlier results from the surface biotinylation experiments.

### 3.2. Rab4A-Positive Tubular Emanations Overlap Well with KDELR-Containing Tubular Emanations from the Golgi

As it is possible that ACBD3 depletion somehow stimulates Rab4A-, Rab14A- and Rab11A-dependent KDELR recycling pathways for the surface-expressed pool of the receptor [11], leading to the increased expression of KDELR at the PM, we performed live cell imaging to investigate whether the export of Golgi-localized KDELR is influenced by ACBD3 depletion. To this end, we co-transfected either the control HT1080 cells or stable ACBD3 KD HT1080 cells (using shRNA lentivirus) with KDELR1 fused to photoactivatable GFP (PA-GFP) and mCherry-Rab4A (Figure 3A,B, Appendix A), mCherry-Rab14A (Figure 3C,D, Appendix A) and mCherry-Rab11A (Appendix A), respectively. We then used a 405 nm laser to activate the Golgi-localized pool of KDELR1-PA-GFP and track the formation of Golgi-derived, KDELR1-positive tubulovesicular carriers using live imaging protocol, as described in the Methods.

Upon examination using live cell confocal imaging, we found greatly a increased frequency of KDELR1-positive tubulovesicular carrier formation in ACBD3 KD cells, compared to the control cells (Figure 3E) [6]. Unexpectedly, Golgi-derived KDELR1-positive tubular carriers showed a relatively poor overlap with both mCherry-Rab14A or Rab11A (Figure 3E), indicating that the increased targeting of KDELR1 to the PM in ACBD3-depleted cells may not result from the elevated rate of Rab14A- and Rab11A-mediated KDELR recycling to the PM. Instead, Golgi-derived KDELR1-positive tubular carriers overlapped very well with mCherry-Rab4A (Figure 3A,B,E), suggesting that KDELR1 export from the Golgi may be facilitated by Rab4A.

### 3.3. Rab4A Co-Localizes with Golgi Markers and Mediates KDELR-Positive, Golgi-Derived Tubular Carrier Formation

Rab4A had been shown to primarily mediate fast recycling pathways at the early endosomes for various surface-expressed receptors [26,27,28,29], but not to mediate receptor trafficking at the Golgi. Interestingly, Rab4A also orchestrates Arl1-dependent BIG1/BIG2 recruitment and the formation of the clathrin adaptor (AP-1, AP-3 and GGA-3)-enriched endosomal sorting domain [30], which had been known to predominantly occur at the TGN [31,32,33,34,35,36]. In order to find out whether Rab4A is associated with the Golgi membrane, we examined the co-localization of EGFP-Rab4A with Golgi marker GM130 and early endosome marker EEA1 using confocal microscopy. The results showed that EGFP-Rab4A co-localized well with EEA1 as reported and a perinuclear pool of EGFP-Rab4A co-localized well with GM130 in HeLa cells. We then examined the co-localization of EGFP-Rab4A with ACBD3 and a *trans*-Golgi network marker TGN46 using confocal microscopy. The results showed that a perinuclear pool of EGFP-Rab4A co-localized well with both ACBD3 and TGN46 in HeLa cells, suggesting that a certain fraction of Rab4A may localize to the TGN and be involved in Golgi-derived tubular carrier formation (Figure 4A–D).

As Rab4A seems to be playing an important role in KDELR export from the Golgi, we then posited that the exogenous over-expression of constitutively active Rab4A may stimulate the surface targeting of KDELR1. To focus on the Golgi-associated pool of KDELR and Rab4A, HT1080 cells were transiently co-transfected overnight with KDELR-PA-GFP and mCherry-Rab4A-Q72L, followed by photoactivation using a 405 nm laser and live imaging protocol, as described in the Methods. The results showed that a significant fraction of Golgi-derived KDELR tubules overlapped with mCherry-Rab4A, suggesting that Rab4A-Q72L expression stimulates KDELR export from the Golgi, rather than at the early endosome (Figure 4E,F, Appendix A).

To examine whether Golgi-derived Rab4A tubules induced in ACBD3 depleted cells actually detach from the Golgi complex to function as a cargo carrier, we transfected HeLa WT or ACBD3-KO cells with mCherry-Rab4A and a Golgi resident protein, β(1, 4)-galactosyltransferase fused to GFP (GT-GFP), overnight. We performed live cell imaging of mCherry-Rab4A and GT-GFP using a high-resolution airyscan confocal microscopy. We found a greatly increased frequency of mCherry-Rab4A-positive tubulovesicular carrier formation from the Golgi in ACBD3-KO cells, compared to the WT cells (Figure 4G,H, Appendix A), suggesting that ACBD3 depletion stimulates Rab4A-positive tubules that detach from the Golgi complex.

### 3.4. Constitutively Active Rab4A-Q72L Greatly Increases Surface Expression of KDELR, Whereas Expression of GDP-Locked Rab4A Mutant or Rab4A Depletion Inhibits KDELR Expression on the Cell Surface

To further confirm these results, we then transfected HT1080 cells with either mCherry-Rab4A-Q72L or S27N (GDP-locked mutant) overnight and checked whether KDELR1 trafficking to the PM may be influenced by the Rab4A mutant expression. The results indeed indicated that Rab4A-Q72L mutant expression greatly increases KDELR1 expression at the PM, whereas S27N mutant failed to do so (Figure 5A), as shown by the surface biotinylation experiments. It is well known that Rab6A and Rab8A are small GTPases involved in the transport of post-Golgi vesicles [37,38,39,40]. We also tested whether the surface level of KDELR1-mCherry was changed after overexpressing Rab6A-Q72L/T27N and Rab8A-Q67L/T22N mutant pairs. The results show that both Rab6A and Rab8A play no role in the surface expression of KDELR1 (Appendix A). As controls, we also performed similar experiments with Rab11A-Q70L/S25N and Rab14A-Q70L/S25N mutant pairs, but did not see any meaningful changes in the surface expression of KDELR1 in these cells (Figure 5B,C), which helped us exclude the possibility of any potential misinterpretation from Rab-cascades [41].

In an effort to obtain a more quantitative measure of KDELR expression on the cell surface, HT1080 cells were co-transfected with lumenally FLAG-tagged KDELR1-mCherry and EGFP-Rab4A-Q72L or Rab4A-S27N mutants, followed by indirect staining using anti-Flag antibody. Upon examination using a confocal microscope, we found that anti-FLAG tag staining for surface-expressed KDELR was readily observed in HeLa-S3 cells transfected with Rab4A-Q72L, but not detected for the cells transfected with Rab4A-S27N or Rab7-Q67L (a control; N = 40) (Figure 5D,E), again confirming that Rab4A is one of the major regulators of KDELR1 trafficking from the Golgi to the PM.

We then asked whether GDP-locked Rab4A mutant or Rab4A depletion by siRNA can inhibit KDELR trafficking to the cell surface. To this end, HT1080 cells expressing KDELR1-mCherry were transfected with EGFP-Rab4A-S27N mutant or Rab4A siRNA, followed by the surface biotinylation protocol to measure its influence on KDELR expression at the PM. Strikingly, the expression of the Rab4A S27N mutant or Rab4A depletion alone significantly reduced the basal surface expression of KDELR (Figure 5F,G; lane 2–3). In cells depleted of ACBD3, the surface expression of KDELR was also reduced to a similar extent by Rab4A-S27N expression or Rab4A depletion (Figure 5F,G; lane 5–6). Taken together, these results suggested that the Rab4A-mediated regulation of KDELR trafficking most likely works downstream of ACBD3.

## 4. Discussion

We had previously shown that surface-expressed KDELR undergoes clathrin-mediated endocytosis and highly complex recycling pathways through the endo-lysosomal system, including Rab11A- and Rab14A-positive recycling endosomes [11]. In that study, we also found that the internalized pool of surface-expressed KDELR does not utilize the Rab4A-dependent fast recycling pathway through the early endosomes. Therefore, it is interesting to note that the Golgi exit of KDELR preferentially uses Rab4A-dependent tubular carriers over the Rab11A- and Rab14A-dependent mechanism. Co-localization analysis using Golgi markers (GM130 and TGN46; Figure 4A–C) and live cell imaging results using KDELR-PA-GFP and mCherry-Rab4A-Q72L (Figure 4D) indicate that (i) there is a small subset of the Rab4A pool associated with the Golgi membranes; (ii) constitutively active Rab4A-Q72L expression induces frequent KDELR-positive tubular emanations from the Golgi, instead of the early endosomes, as had been widely accepted previously [26,30,42]; and (iii) the knockdown of Rab4A inhibits formation of Golgi-derived tubular emanation in ACBD3-depleted cells.

While we cannot exclude a possibility that Rab4A-Q72L expression could accelerate the fast recycling route of internalized surface-expressed KDELR, leading to increased KDELR expression on the PM, our live imaging results suggest that the Rab4A-mediated export of Golgi-localized KDELR to the PM is also likely to contribute directly to increased KDELR expression on the cell surface.

In addition, as both Rab11A-Q70L and Rab14A-Q70L expression had negligible influence on the surface expression of KDELR (Figure 5B,C), there is little reason to believe that Rab4A-Q72L-induced tubular-carrier-containing KDELR (Figure 4D) may be derived from the perinuclear recycling endosomes, instead of the Golgi. Thus, we conclude that Rab4A may mediate the Golgi exit of certain recycling receptors at the TGN (Figure 6), which has been overlooked so far [26,27,43,44,45].

How does ACBD3 regulate the surface expression of KDELR? Since ACBD3 control KDELR localization to the Golgi under a steady-state condition by modulating its interaction with PKA [6], we initially hypothesized that PKA (or the activation of related Src kinase-dependent signaling [46]) may dictate KDELR’s exit from the TGN. Unexpectedly, however, we found that either the expression of PKA Cα or expression of Src-E381G-myc had a negligible effect on the surface expression of KDELR (Figure 1B,C). Overall, it is not yet clear if other unknown KDEL cargo-induced signaling pathways may exist and regulate the Golgi exit of KDELR.

It is possible that the physical association between KDELR and ACBD3 may exert an inhibitory influence, effectively minimizing Golgi leakage of KDELR under a steady-state condition. Since ACBD3 is known to localize throughout the Golgi, including the TGN [47,48,49], this is a reasonable speculation, although the mechanism of KDELR release from ACBD3 is currently poorly understood.

Alternatively, since ACBD3 depletion greatly increases Arf1-positive tubular carrier formation [6], we suspect that increased Arf1 activity in ACBD3-depleted cells may partly explain increased KDELR export from the Golgi, although this requires further in-depth study.

Recently, KDELR was shown to promote FAK recruitment and activation to the area of ECM degradation in invadopodia and also stimulate directed vesicular trafficking and protrusive membrane dynamics at the PM [50,51]. Moreover, secreted ER chaperones, such as ERp57 and GRP78, were reported to stimulate cancer cell migration and invasion, in which the latter was shown to promote ECM degradation and FAK activation as well [52,53,54]. Thus, it would be intriguing to investigate whether surface-expressed KDELR may interact with secreted ER-chaperones and facilitate the invasive behavior of various cancers. In this context, it is worth noting that ACBD3, Arf1, PI4Kβ and Rab4A all have been implicated in breast cancer progression and prognosis and found to be coordinately regulated in the same region of chromosome 1q [55,56,57,58,59,60].

## 5. Conclusions

Since its discovery in the early 1990s, KDELR has been extensively studied for its role in the retrograde transport of leaked ER chaperones from the Golgi. Recently, studies have shown that KDELR might also play additional roles as a cell surface receptor, although exactly how KDELR exit the Golgi and travel to the plasma membranes had remained elusive. Our new results shed light on the previously unknown mechanism for the post-Golgi trafficking of KDELR to the cell surface, which could play a role in promoting secreted ER-chaperone-induced cell migration and the invasion of cancer cells. While we identified two regulatory factors, ACBD3 and Rab4, in this study, future study is required to further characterize the molecular composition of the membrane microdomains responsible for KDELR export from the TGN and the specific signaling events that lead to increased KDELR trafficking to the cell surface during oncogenic transformation or embryonic development.

## Figures and Tables

**Figure 1 cells-12-01079-f001:**
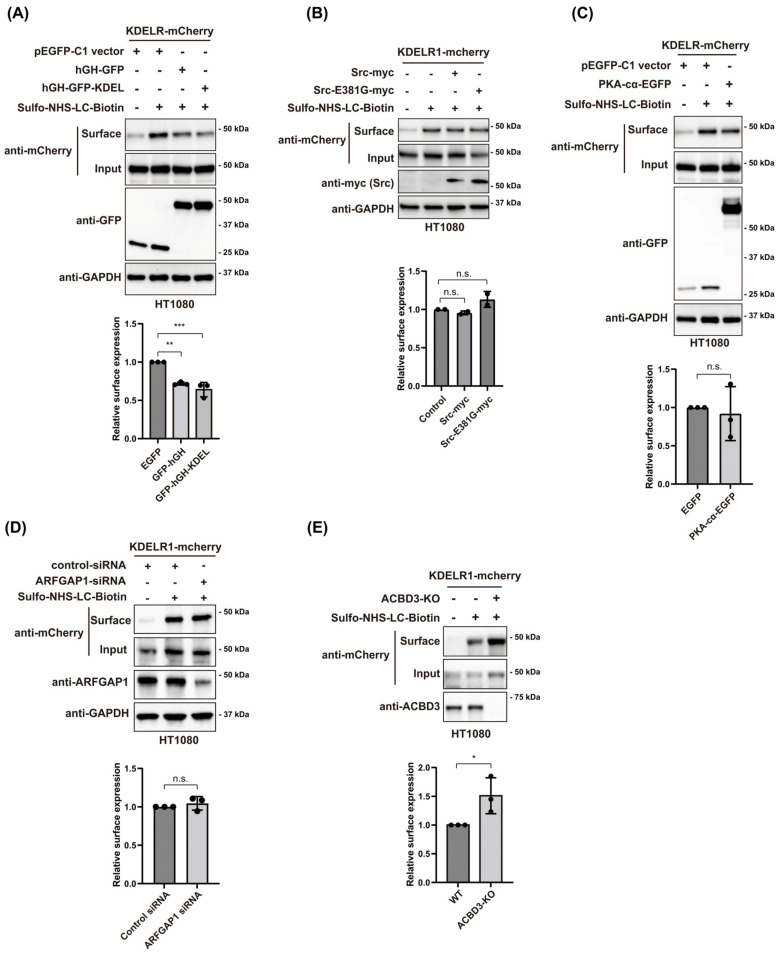
A small scale screening of regulators of KDELR transport to the plasma membrane. (**A**) Expression of hGH-GFP-KDEL causes decreased surface expression of KDELR1-mCherry compared to GFP or hGH-GFP control. HT1080 cells were co-transfected with KDEL-R1-mCherry and hGH-GFP-KDEL, hGH-GFP or GFP control and cell surface expression of KDEL-R1-mCherry was detected via cell surface biotinylation. The normalized cell surface level of KDEL-R-mCherry using densitometric analysis is shown at the bottom. Statistical analysis was performed using one-way ANOVA with a Tukey’s post hoc test (mean ± SD; **, *p* < 0.01; ***, *p* < 0.001). (**B**) HT1080 were co-transfected with KDEL-R1-mCherry and Src-WT or a constitutive active mutant E381G, followed by cell surface biotinylation. Biotinylated proteins were isolated using streptavidin-agarose and subjected to Western blot analysis using the indicated antibodies. The normalized cell surface level of KDEL-R-mCherry using densitometric analysis is shown at the bottom. Statistical analysis was performed using one-way ANOVA with a Tukey’s post hoc test (mean ± SD; n.s., not significant). (**C**) HT1080 cells co-transfected with KDEL-R1-mCherry and PKA-Cα-EGFP or EGFP control and cell surface expression of KDEL-R1-mCherry was detected via cell surface biotinylation. Normalized cell surface level of KDEL-R-mCherry using densitometric analysis is shown at the bottom. Statistical analysis was performed using Student’s *t*-test (mean ± SD; n.s., not significant). (**D**) HT1080 cells were transfected with ARFGAP1 siRNA for 48h, and then transfected with KDEL-R1-mCherry for 18 h, followed by cell surface biotinylation. Biotinylated proteins were isolated using streptavidin-agarose and subjected to Western blot analysis using the indicated antibodies. The normalized cell surface level of KDEL-R-mCherry using densitometric analysis is shown at the bottom. Statistical analysis was performed using Student’s *t*-test (mean ± SD; n.s., not significant). (**E**) WT or ACBD3-KO HT1080 cells were transfected with KDEL-R1-mCherry for 18 h, followed by cell surface biotinylation. Biotinylated proteins were isolated using streptavidin-agarose and subjected to Western blot analysis using the indicated antibodies. The normalized cell surface level of KDEL-R-mCherry using densitometric analysis is shown at the bottom. Statistical analysis was performed using Student’s *t*-test (mean ± SD; *, *p* < 0.05).

**Figure 2 cells-12-01079-f002:**
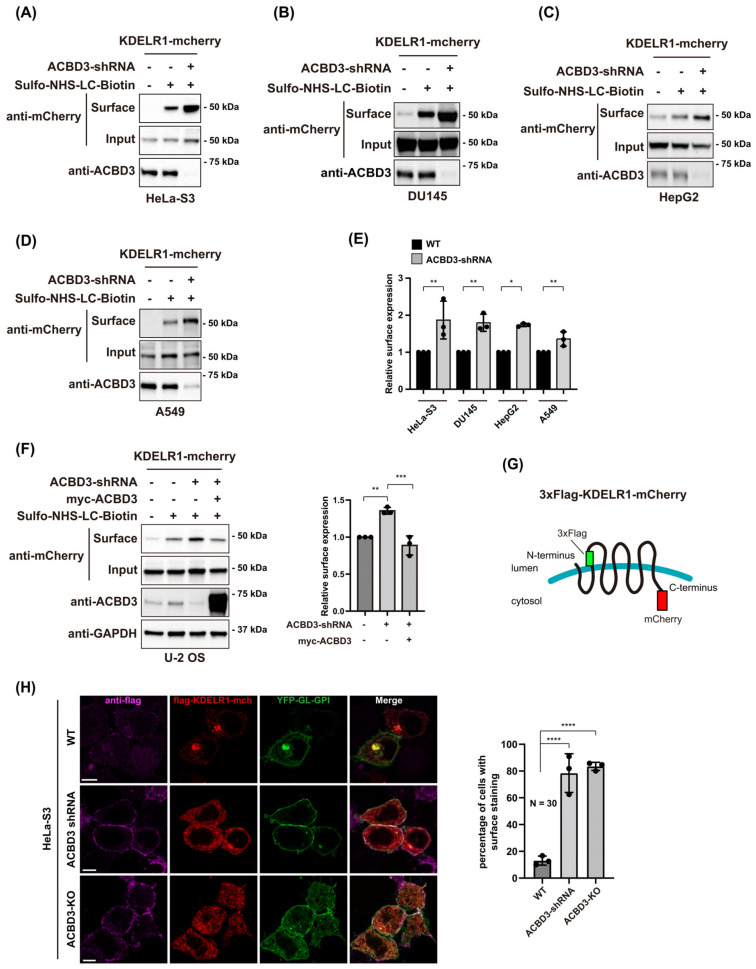
ACBD3 depletion causes significantly increased expression of KDEL-R1 on the cell surface. (**A**–**D**) Cell surface expression of KDEL-R1-mCherry is significantly increased in a number of ACBD3 depleted cell lines, compared to the control cells. Cell surface expression of KDEL-R1-mCherry was probed via cell surface biotinylation protocol using sulfo-NHS-LC-biotin. We used a shRNA/lentiviral transduction method to establish four mammalian cell lines, for which ACBD3 was stably knocked down, as described in the Methods. Briefly, WT or ACBD3 depleted cells were transfected with KDEL-R1-mCherry for 18 h, followed by cell surface biotinylation. Biotinylated proteins were isolated using streptavidin-agarose and subjected to Western blot analysis using the indicated antibodies. (**E**) Bar graphs showing normalized cell surface expression of KDEL-R-mCherry in various cell lines using densitometric analysis. Statistical analysis was performed using one-way ANOVA with a Tukey’s post hoc test (mean ± SD; *, *p* < 0.05; **, *p* < 0.01). (**F**) Cell surface expression of KDEL-R1-mCherry is significantly increased in ACBD3 depleted U-2 OS cells, compared to the control cells, which could be restored by the exogenous expression of RNAi-resistant myc-ACBD3. The normalized cell surface level of KDEL-R-mCherry using densitometric analysis is shown on the right. Statistical analysis was performed using one-way ANOVA with a Tukey’s post hoc test (mean ± SD; **, *p* < 0.01; ***, *p* < 0.001) (**G**) Schematic representation of 3xFLAG-KDEL-R1-mCherry. A 3xFLAG tag was inserted into the first luminal (extracellular) loop of KDEL-R1. (**H**) Increased cell surface staining of 3xFLAG-KDEL-R1 observed via confocal microscopy in ACBD3-knockdown/knockout HeLa S3 cells. Control or ACBD3-knockdown/knockout HeLa S3 cells were co-transfected with 3xFLAG-KDEL-R1-mCherry and YFP-GL-GPI, which served as a cell surface marker, for 18 h. The living cells were stained using anti-FLAG tag antibody at 4 °C and fixed with 4% paraformaldehyde, followed by staining with the secondary antibodies. Scale bars = 10 µm. Bar graphs showing the percentage of cells with surface staining of KDEL-R1 in 3xFLAG-KDEL-R1-mCherry transfected HeLa-S3 cells. N = 30 cells. Statistical analysis was performed using one-way ANOVA with a Tukey’s post hoc test (mean ± SD; ****, *p* < 0.0001).

**Figure 3 cells-12-01079-f003:**
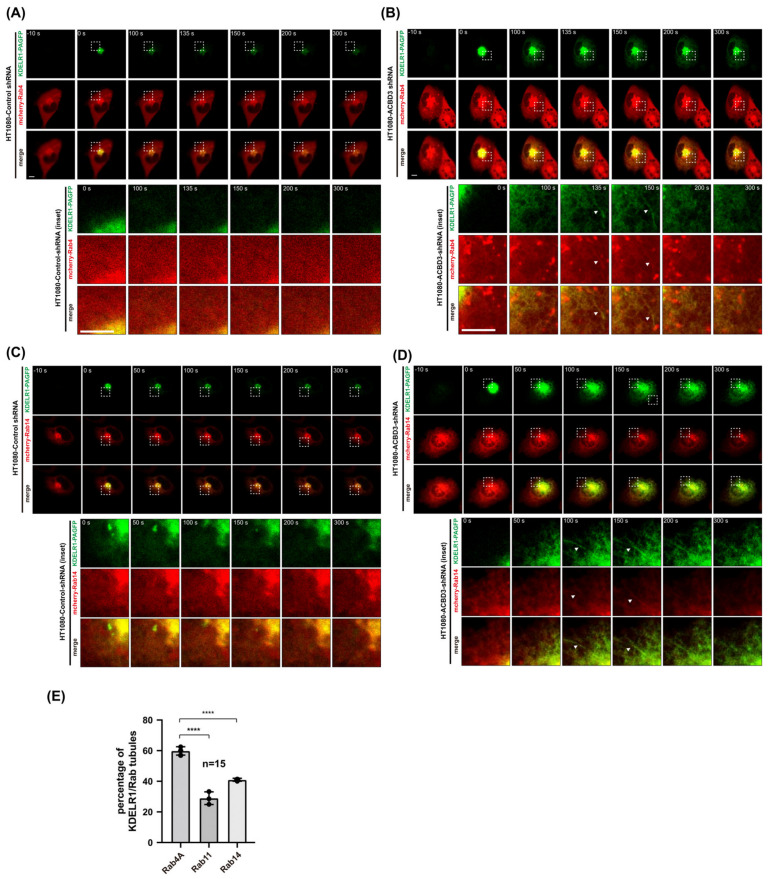
ACBD3 depletion results in increased trafficking of KDEL receptor to the PM via Rab4A-positive tubular carriers at the Golgi. (**A**–**D**) To examine the post-Golgi trafficking itineraries of KDEL receptor to the PM, WT or ACBD3-depleted HT1080 cells were co-transfected with photoactivatable KDELR1-PA-GFP and mCherry-Rab4A/14 plasmids for 18 h. The KDELR1-PA-GFP in the Golgi were then activated by selecting an ROI of the mCherry-Rab4A/14 perinuclear region for intense 405 nm laser irradiation, and the transport out of the Golgi was monitored via live cell imaging acquired every 5 s for 5 min. Imaging sequences prior to photoactivation (−10 s) and immediately after photoactivation (0 s), and the indicated time points following photoactivation, are presented here. Magnified regions of interest (indicated by white boxes) from WT and ACBD3-KO cells at the indicated time points show Golgi-derived tubules which are highlighted by white arrowheads. Scale bars = 5 µm. (**E**) Bar graphs showing the percentage of KDELR1/Rab tubules present in ACBD3-depleted HT1080 cells. *N* = 15 cells. Statistical analysis was performed using one-way ANOVA with a Tukey’s post hoc test (mean ± SD; ****, *p* < 0.0001).

**Figure 4 cells-12-01079-f004:**
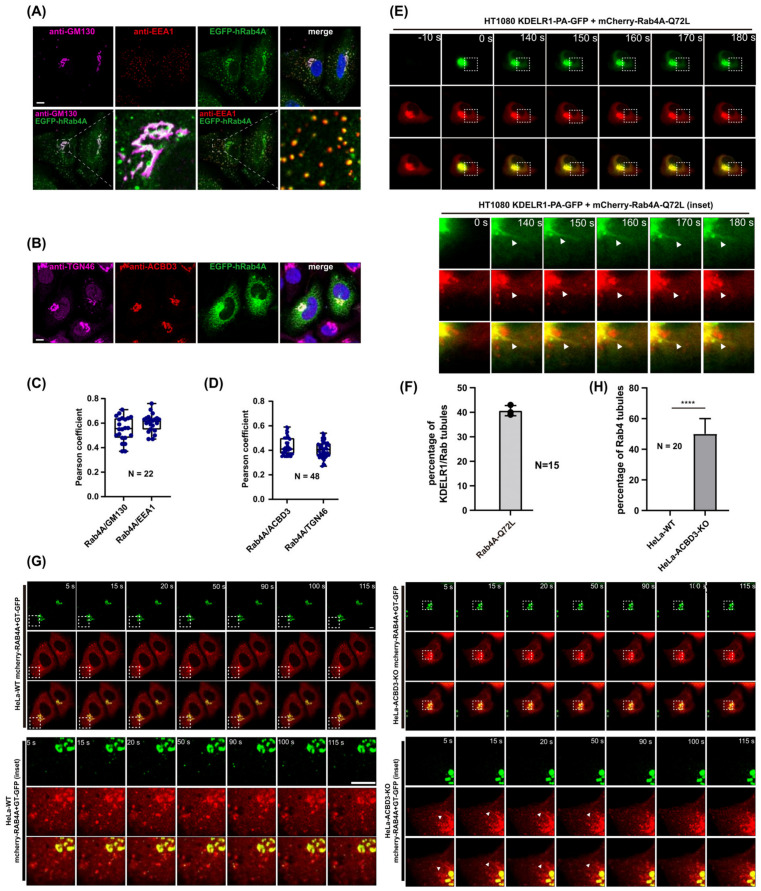
Overexpression of a constitutive active mutant Rab4A-Q72L promotes KDEL-R1 trafficking to the PM via Rab4A-positive tubular carriers at the Golgi. (**A**–**D**) To examine the Golgi localization of Rab4A, HeLa cells were transfected with human EGFP-Rab4A plasmids for 18 h, followed by staining with anti-GM130 (*cis*-Golgi) and anti-EEA1 (early endosome) (**A**) or anti-ACBD3 and TGN46 (*trans*-Golgi network) (**B**). Scale bars = 10 µm. Co-localization (Pearson’s R) was determined. *N* = 22 for (**C**), *N* = 48 for (**D**). (**E**,**F**) To examine the role of Rab4A in promoting KDEL-R1 trafficking to the PM from Golgi, WT HT1080 cells were co-transfected with photoactivatable KDELR1-PA-GFP and mCherry-Rab4A-Q72L plasmids for 18 h. The KDELR1-PA-GFP in the Golgi were then activated by selecting an ROI of the mCherry-Rab4A Golgi region for intense 405 nm laser irradiation and the transport out of the Golgi was monitored by live cell imaging acquired every 5 s for 5 min. Imaging sequences prior to photoactivation (−10 s) and immediately after photoactivation (0 s), and the indicated time points following photoactivation are presented here. Magnified regions of interest (indicated by white boxes) at the indicated time points show Golgi-derived tubules which are highlighted by white arrowheads. Scale bars = 5 µm. Bar graphs showing the percentage of KDELR1/Rab tubules present in HT1080 cells. *N* = 15 cells. (**G**,**H**) HeLa WT or ACBD3-KO cells were transfected with mCherry-Rab4A and GT-GFP, overnight. Live cell imaging of mCherry-Rab4A and GT-GFP were performed using a high-resolution airyscan confocal microscope. Magnified regions of interest (indicated by white boxes) at the indicated time points show Golgi-derived tubules which are highlighted by white arrowheads. Scale bars = 5 µm. Bar graphs showing the percentage of Rab4A tubules present in HeLa WT or ACBD3-KO cells. *N* = 20 cells. Statistical analysis was performed using two-tailed, paired *t*-test (mean ± SD; ****, *p* < 0.0001).

**Figure 5 cells-12-01079-f005:**
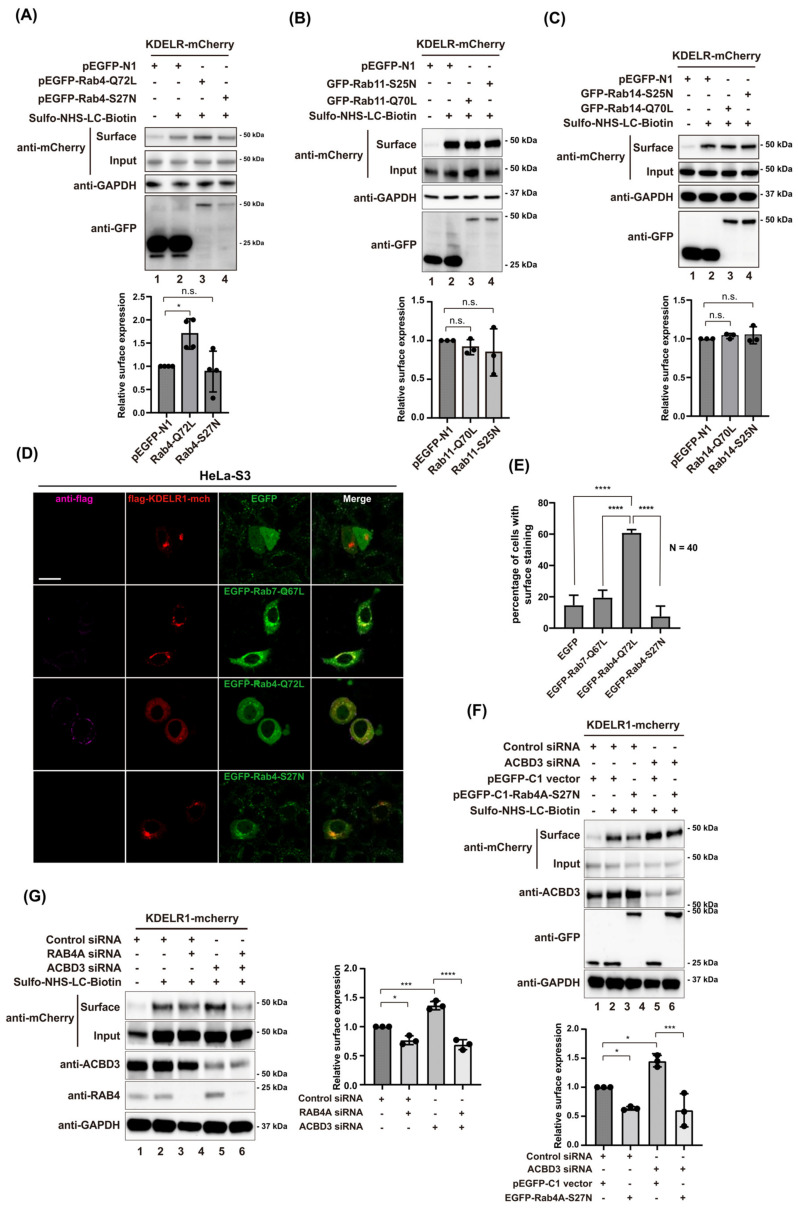
Rab4A plays a crucial role in the cell surface expression of KDEL-R1. (**A**–**C**) The cell surface expression of KDEL-R1-mCherry is significantly increased in HT1080 cells when overexpressing the constitutive active mutant Rab4A-Q72L, but not Rab11A-Q70L or Rab14A-Q70L, compared to the control cells. HT1080 cells were transfected with KDEL-R1-mCherry and indicated Rab plasmids for 18 h, followed by cell surface biotinylation. Biotinylated proteins were isolated using streptavidin-agarose and subjected to Western blot analysis using the indicated antibodies. The normalized cell surface level of KDEL-R-mCherry using densitometric analysis is shown at the bottom. Statistical analysis was performed using one-way ANOVA with a Tukey’s post hoc test (mean ± SD; n.s., not significant; *, *p* < 0.05) (**D**) Increased cell surface staining of 3xFLAG-KDEL-R1 observed via confocal microscopy in HeLa S3 cells expressing constitutive active mutant Rab4A-Q72L, compared to the cells expressing pEGFP-C1 vector, EGFP-Rab7-Q67L or EGFP-Rab4A-S27N. HeLa S3 cells were co-transfected with 3xFLAG-KDEL-R1-mCherry and pEGFP-C1 vector or indicated Rab plasmids for 18 h. The living cells were stained by anti-FLAG tag antibody at 4 °C and fixed with 4% paraformaldehyde, followed by staining with the secondary antibodies. Scale bars = 5 µm. (**E**) Bar graphs showing the percentage of cells with surface staining of KDEL-R1 in 3xFLAG-KDEL-R1-mCherry transfected HeLa-S3 cells. *N* = 40 cells. Statistical analysis was performed using one-way ANOVA with a Tukey’s post hoc test (mean ± SD; ****, *p* < 0.0001). (**F**) Cell surface expression of KDEL-R1-mCherry is decreased in both HT1080 WT or ACBD3 knockdown cells when overexpressing a dominant negative mutant Rab4A-S27N. HT1080 cells were transfected with control siRNA or ACBD3 siRNA for 48 h. Then, cells were transfected with KDEL-R1-mCherry and EGFP-Rab4A-S27N or pEGFP-C1 vector plasmids for 18 h, followed by cell surface biotinylation. Biotinylated proteins were isolated using streptavidin-agarose and subjected to Western blot analysis using the indicated antibodies. The normalized cell surface level of KDEL-R-mCherry using densitometric analysis is shown at the bottom. Statistical analysis was performed using one-way ANOVA with a Tukey’s post hoc test (mean ± SD; *, *p* < 0.05; ***, *p* < 0.001). (**G**) Cell surface expression of KDEL-R1-mCherry is decreased in both HT1080 WT or ACBD3 knockdown cells when transfected with Rab4A siRNA. HT1080 cells were transfected with Rab4A and control siRNA or Rab4A and ACBD3 siRNA for 48h. Then, cells were transfected with KDEL-R1-mCherry for 18 h, followed by cell surface biotinylation. Biotinylated proteins were isolated using streptavidin-agarose and subjected to Western blot analysis using the indicated antibodies. The normalized cell surface level of KDEL-R-mCherry using densitometric analysis is shown at the bottom. Statistical analysis was performed using one-way ANOVA with a Tukey’s post hoc test (mean ± SD; *, *p* < 0.05; ***, *p* < 0.001; ****, *p* < 0.0001).

**Figure 6 cells-12-01079-f006:**
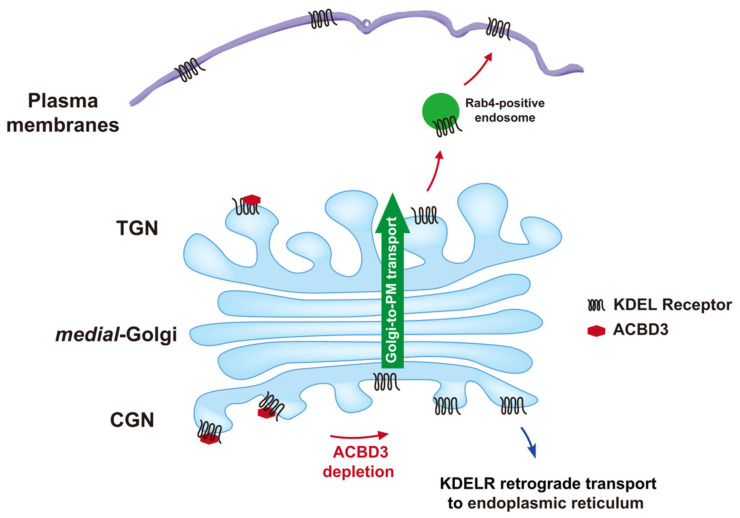
Schematic diagram depicting the proposed role of ACBD3 and Rab4A in controlling KDELR trafficking to the PM.

## Data Availability

All data generated or analyzed during this study are included in this published article and its Appendix A.

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
