# Peer review of "KDEL Receptor Trafficking to the Plasma Membrane Is Regulated by ACBD3 and Rab4A-GTP"

_cells, 2023, doi:10.3390/cells12071079_

Round 1

Reviewer 1 Report

In this manuscript, the authors have investigated how a fraction of the KDEL receptor (KDELR) is transported to the plasma membrane. They show that the depletion of ABCD3, previously shown by the same group to be involved in the retention of KDELR in the Golgi apparatus, increases cell surface expression of KDELR. They also provide evidence that RAB4 is involved in the transport of KDELR from the Golgi complex to the plasma membrane.

This is an interesting study that shed some light of the trafficking of KDELR and the role of ABCD3. A surprising result, and the most original one, is the involvement of RAB4, whose role in recycling events from early endosomes is well established, in post-Golgi transport of KDELR. Unfortunately, this lacks strong evidence and should be better investigated.

Major points:

1)    Numerous studies have documented the association of RAB4 with early endosomes. In this study, the authors found that a large pool of overexpressed RAB4A co-localizes with Golgi markers (Fig. 4A-C). What is puzzling is that RAB4 staining appears to be only visible in the Golgi area whereas early endosomes are usually located at the cell periphery. What is the explanation for this observation? Co-localization of GFP-RAB4 with endosomal markers (RAB5, EEA1,..) should be provided.

2)    The experiments showing the presence of RAB4 in Golgi-derived KDELR-positive tubules are far from being convincing (Fig. 4D). I see no evidence that these tubules detach from the Golgi complex, which would be expected in they correspond to transport carriers. How many tubules per Golgi are we talking about? The corresponding videos should be provided.

3)    The present results do not allow to exclude the possibility that the increase in cell surface expression of KDELR in cells overexpressing activated RAB4 (Q72L) is due to an effect on internalization/recycling of cell surface associated KDELR. The authors exclude this possibility based on a previously published work (Jia et al, Cel Mol Life Sci 2021). Unfortunately, this publication is not in open access and I could not read it. To strengthen the role of RAB4 in post-Golgi trafficking of KDELR, the authors should investigate whether the RAB GTPases known to regulate post-Golgi transport (i.e. RAB6 and RAB8) could be involved.

4)    The role of RAB4 was investigated by overexpressing dominant-active and dominant-negative forms of the protein. Although this approach has been extensively used to study the function of RAB GTPases, it may lead to artefacts to the fact that these proteins share common effectors (for instance RAB4 and RAB5). Some experiments should be repeated by depleting RAB4 with si or sh RNAs.

Minor comments:

- Fig. 2H: Flag (KDELR) staining is barely visible in ACBD3-KO cells

- Fig. 5D-F: Flag staining is also barely visible in cells overexpressing RAB4 constructs. These experiments have been quantified by counting the number of positive cells. FACS analysis would be more quantitative. The KDELR staining looks quite different in cells overexpressing RAB7 Q67L. Is there a reason for that?

- Two RAB4 isoforms exist, RAB4A and RAB4B. I guess that the experiments have been performed with RAB4A (see Fig. 4A-C). This should be stated more clearly, i.e. by replacing RAB4 by RAB4A throughout the text and Figures.

- The original papers describing the localization of RAB4 and its role in recycling should be cited (van der Sluijs et al, PNAS 1991, Cell 1992).

Author Response

Dear Editor and reviewers

Thank you for your great input and criticism of our work. We tried our best to address the reviewers’ concerns and questions through the hard work during the past three months or so. We hope that this revision is acceptable for the journal’s high standard for publication.

Reviewer#1:

In this manuscript, the authors have investigated how a fraction of the KDEL receptor (KDELR) is transported to the plasma membrane. They show that the depletion of ABCD3, previously shown by the same group to be involved in the retention of KDELR in the Golgi apparatus, increases cell surface expression of KDELR. They also provide evidence that RAB4 is involved in the transport of KDELR from the Golgi complex to the plasma membrane.

This is an interesting study that shed some light of the trafficking of KDELR and the role of ABCD3. A surprising result, and the most original one, is the involvement of RAB4, whose role in recycling events from early endosomes is well established, in post-Golgi transport of KDELR. Unfortunately, this lacks strong evidence and should be better investigated.

Major points:

1)    Numerous studies have documented the association of RAB4 with early endosomes. In this study, the authors found that a large pool of overexpressed RAB4A co-localizes with Golgi markers (Fig. 4A-C). What is puzzling is that RAB4 staining appears to be only visible in the Golgi area whereas early endosomes are usually located at the cell periphery. What is the explanation for this observation? Co-localization of GFP-RAB4 with endosomal markers (RAB5, EEA1,..) should be provided.

A: Thank you for raising an important question. We re-did this figure by overexpression of EGFP-Rab4A in HeLa cells. We co-stained the cells with anti-GM130 (Golgi marker) and anti-EEA1 (endosome marker) or anti-ACBD3 (Golgi) and anti-TGN46 (trans-Golgi network). We found that EGFP-Rab4A co-localized with both GM130 and EEA1 well (Figure 4A-D ). We also found that a fraction of Rab4 may localize to the TGN and be involved in Golgi-derived tubular carrier formation.

2)    The experiments showing the presence of RAB4 in Golgi-derived KDELR-positive tubules are far from being convincing (Fig. 4D). I see no evidence that these tubules detach from the Golgi complex, which would be expected in they correspond to transport carriers. How many tubules per Golgi are we talking about? The corresponding videos should be provided.

A: Thank you for raising this important question. To clearly show the Rab4A tubules detach from the Golgi complex, we transfected HeLa WT or ACBD3-KO cells with mCherry-Rab4A and a Golgi resident protein, β(1, 4)-galactosyltransferase fused to GFP (GT-GFP), overnight. We performed live cell imaging of mCherry-Rab4A and GT-GFP using a high-resolution airyscan confocal. Again, we found greatly increased frequency in mCherry-Rab4A-positive tubulovesicular carrier formation from the Golgi in ACBD3-KO cells, compared to the WT cells (Figure 4G-H, Video S8-9). Although this was technically challenging experiments, we observed upon closer inspection of the live imaging results that these Rab4-positive tubules do detach from the Golgi complex, which moved away from the Golgi complex toward the cell periphery.

3)    The present results do not allow to exclude the possibility that the increase in cell surface expression of KDELR in cells overexpressing activated RAB4 (Q72L) is due to an effect on internalization/recycling of cell surface associated KDELR. The authors exclude this possibility based on a previously published work (Jia et al, Cel Mol Life Sci 2021). Unfortunately, this publication is not in open access and I could not read it. To strengthen the role of RAB4 in post-Golgi trafficking of KDELR, the authors should investigate whether the RAB GTPases known to regulate post-Golgi transport (i.e. RAB6 and RAB8) could be involved.

A: Thank you for raising this important question. Based on your suggestion, we tested the role of Rab6 and Rab8 in influencing surface level of KDELR1-mCherry. The results show that both Rab6 and Rab8 play no role in the surface expression of KDELR1 (supplementary Figure 2A-B).

4)    The role of RAB4 was investigated by overexpressing dominant-active and dominant-negative forms of the protein. Although this approach has been extensively used to study the function of RAB GTPases, it may lead to artefacts to the fact that these proteins share common effectors (for instance RAB4 and RAB5). Some experiments should be repeated by depleting RAB4 with si or sh RNAs.

 A: Thanks for your suggestion. We measured the influence of Rab4A depletion on KDELR expression on the PM. Rab4 depletion alone significantly reduced basal surface expression of KDELR (Figure 5F-G; lane 2-3). In cells depleted of ACBD3, surface expression of KDELR was also reduced to a similar extent (Figure 5F-G; lane 5-6).

Minor comments:

- Fig. 2H: Flag (KDELR) staining is barely visible in ACBD3-KO cells

 A: Thank you for pointing out this. We adjusted the contrast and added the quantified graph.

- Fig. 5D-F: Flag staining is also barely visible in cells overexpressing RAB4 constructs. These experiments have been quantified by counting the number of positive cells. FACS analysis would be more quantitative. The KDELR staining looks quite different in cells overexpressing RAB7 Q67L. Is there a reason for that?

 A: Thank you for pointing out this. We adjusted the contrast and the figure looks clear now. FACS analysis is a good way to do quantification. But it may not suitable for KDELR because of its low surface expression level. For the Rab7 Q67L overexpressing group, we replaced with a new confocal image.

- Two RAB4 isoforms exist, RAB4A and RAB4B. I guess that the experiments have been performed with RAB4A (see Fig. 4A-C). This should be stated more clearly, i.e. by replacing RAB4 by RAB4A throughout the text and Figures.

 A: Thank you for pointing out this. We revised the text.

- The original papers describing the localization of RAB4 and its role in recycling should be cited (van der Sluijs et al, PNAS 1991, Cell 1992).

 A: Thank you for pointing out this. We referred these studies.

Reviewer 2 Report

Major comments

Previously the authors described the function of ACBD3 as a negative regulator of PKA signaling. They reported on BMC the depletion of ACBD3 which induces KDELR relocation from Golgi to ER. Now the authors described a new function of ACBD3 as a regulator of KDELR transport to the plasma membrane. The authors mentioned the previous article, but they ignore the KDELR relocation to the ER under ACBD3 depletion reported by themselves previously. Instead, they make a statement regarding ACBD3 role in Golgi localization that is true but lacks to include the ER.

A general comment regards to the blots throughout the manuscript should be quantified, including statistical analysis also. The input lane appears quite variable, so the level of mcherry overexpression might affect the interpretation of the results. The authors might calculate surface/total KDELR to compare the effect among the treatments.

For example, “Both Src and PKA have been shown to 85 regulate KDELR retrograde transport to the ER. Thus, we studied the surface expression of 86 KDELR1-mCherry while overexpressing a constitutively active mutant Src(E381G) or PKA- 87 Cα. The results showed that both Src and PKA expression do not affect surface expression of KDELR1-mCherry, compared to the control (Figure 1B&C)”. How the authors may have been sure about this statement without quantitative analysis of the blots. Instead, from the author’s interpretation, I can observe a less biotinylated mcherry-KDELR by PKAc overexpression in figure 1C. The same is for figure 1B which shows a high surface/input ratio. This change completely the interpretation of the results.

Figure 1, The authors detect mcherry-KDELR at the cell surface without Sulfo-NHS-LC-Biotin that appears in Figures 1 A, B, C, and D and absent in E (line 1 for each blot). I guess they perform first Biotin Ipp and then mcherry detection by blot. So, should not be able to get any mcherry signal by blot in the absence of Sulfo-NHS-LC-Biotin.

Figure 2, To compare the results obtained by blot and IFI the authors should use the same KDELR construct. Here, by fluoresce is clear that KDELR goes back to the ER under ACBD3 depletion. Just a very small fraction goes apparently to the cell surface.

Figure 3, the images are extremely small and the tubules that the authors claim are quite dim structures that are not clear enough to be part of the Golgi or ER tubules, endosomes, or even plasma membrane tubulation. The images are not good enough to make any conclusion.

Figure 4, The rab4 tubules are not visible and the role of rab4 is not clear regarding its effect on neither retro nor anterograde KDELR transport. Moreover, as the authors mentioned rab4 has been involved in fast recycling to the plasma membrane from endosomes, where has been detected endogenous and overexpressed rab4. In the author’s hands, endosomal localization is not visible, this is quite strange. Also, the Golgi localization that the authors claim should be performed by super-resolution microscopy (SIM) has previously been done. This is because in some cell lines endosomes are located very close to the Golgi and look like Golgi. In 4D the y-axis on the bar plot should be replaced by a fraction or shown as a percentage.

Figure 5, the great increase in KDELR at the cell surface obtained by blot it is not visible by IFI. Once again, the amount of plasma membrane KDELR is quite variable among the experiments. Should be performed a surface/total ratio and statistical analysis. The plasma membrane signal detected by IFI is quite dim.

Minor comments

In lines 60 and 62 there are some missing spaces.

In lines 85-86 there are 2 missing references

Line 91 a missing space

Author Response

Dear Editor and reviewers

Thank you for your great input and criticism of our work. We tried our best to address the reviewers’ concerns and questions through the hard work during the past three months or so. We hope that this revision is acceptable for the journal’s high standard for publication.

Reviewer#2: Major comments

Previously the authors described the function of ACBD3 as a negative regulator of PKA signaling. They reported on BMC the depletion of ACBD3 which induces KDELR relocation from Golgi to ER. Now the authors described a new function of ACBD3 as a regulator of KDELR transport to the plasma membrane. The authors mentioned the previous article, but they ignore the KDELR relocation to the ER under ACBD3 depletion reported by themselves previously. Instead, they make a statement regarding ACBD3 role in Golgi localization that is true but lacks to include the ER.

A general comment regards to the blots throughout the manuscript should be quantified, including statistical analysis also. The input lane appears quite variable, so the level of mcherry overexpression might affect the interpretation of the results. The authors might calculate surface/total KDELR to compare the effect among the treatments.

For example, “Both Src and PKA have been shown to 85 regulate KDELR retrograde transport to the ER. Thus, we studied the surface expression of 86 KDELR1-mCherry while overexpressing a constitutively active mutant Src(E381G) or PKA- 87 Cα. The results showed that both Src and PKA expression do not affect surface expression of KDELR1-mCherry, compared to the control (Figure 1B&C)”. How the authors may have been sure about this statement without quantitative analysis of the blots. Instead, from the author’s interpretation, I can observe a less biotinylated mcherry-KDELR by PKAc overexpression in figure 1C. The same is for figure 1B which shows a high surface/input ratio. This change completely the interpretation of the results.

A: Thank you for raising this important question. We calculated the surface/total KDELR and showed the bar graph in the figures. We also revised the text according to the quantification.

Figure 1, The authors detect mcherry-KDELR at the cell surface without Sulfo-NHS-LC-Biotin that appears in Figures 1 A, B, C, and D and absent in E (line 1 for each blot). I guess they perform first Biotin Ipp and then mcherry detection by blot. So, should not be able to get any mcherry signal by blot in the absence of Sulfo-NHS-LC-Biotin.

A: Thank you for raising this important question. The experiments were performed firstly biotin incubation and then pulldown was performed using streptavidin-beads. Sometimes, there was non-specific binding in the no biotin incubated group.

Figure 2, To compare the results obtained by blot and IFI the authors should use the same KDELR construct. Here, by fluoresce is clear that KDELR goes back to the ER under ACBD3 depletion. Just a very small fraction goes apparently to the cell surface.

A: Thank you for pointing out this. The KDELR construct used for blot is C-terminal mCherry tagged KDELR. The one used for IF is to add 3xFlag in the first lumenal loop of KDELR-mCherry construct, in order to stain surface KDELR using anti-Flag antibody. Only a fraction goes to the cell surface.

Figure 3, the images are extremely small and the tubules that the authors claim are quite dim structures that are not clear enough to be part of the Golgi or ER tubules, endosomes, or even plasma membrane tubulation. The images are not good enough to make any conclusion.

A: Thank you for pointing out this. We will submit figures with high dpi. Meanwhile, we also performed live cell imaging using high-resolution airyscan confocal in figure 4G.

Figure 4, The rab4 tubules are not visible and the role of rab4 is not clear regarding its effect on neither retro nor anterograde KDELR transport. Moreover, as the authors mentioned rab4 has been involved in fast recycling to the plasma membrane from endosomes, where has been detected endogenous and overexpressed rab4. In the author’s hands, endosomal localization is not visible, this is quite strange. Also, the Golgi localization that the authors claim should be performed by super-resolution microscopy (SIM) has previously been done. This is because in some cell lines endosomes are located very close to the Golgi and look like Golgi. In 4D the y-axis on the bar plot should be replaced by a fraction or shown as a percentage.

A: Thank you for raising an important question. We changed the cell line to HeLa cells to check the localization of EGFP-Rab4A. We found that EGFP-Rab4A clearly localized to both the Golgi and the early endosome. In HT1080 cells, the endosomes may be located very close to the Golgi, and it’s hard to see clear endosome only localization. We revised the bar graph in figure 4D.

Figure 5, the great increase in KDELR at the cell surface obtained by blot it is not visible by IFI. Once again, the amount of plasma membrane KDELR is quite variable among the experiments. Should be performed a surface/total ratio and statistical analysis. The plasma membrane signal detected by IFI is quite dim.
A: Thank you for raising an important question. The cell surface KDELR obtained by blot is 30% fraction pulldown from a well of a 6-well plate, while IF only shows a single cell. We performed and added the surface/total ratio quantification in the figures. We also adjusted the contrast to show clear surface KDELR in the IF figures.

Minor comments

In lines 60 and 62 there are some missing spaces.

In lines 85-86 there are 2 missing references

Line 91 a missing space

A: Thank you for pointing out these minor mistakes. We revised the text, accordingly.

Round 2

Reviewer 1 Report

The authors have met my main comments/criticisms in a satisfactory way.

Minor points: RAB4 has not been replaced by RAB4A throughout the text; please precise which RAB6 and RAB8 isoforms have been used.

Author Response

Minor points: RAB4 has not been replaced by RAB4A throughout the text; please precise which RAB6 and RAB8 isoforms have been used.

Answer: Thank you for pointing out. We have now revised the text, accordingly.

Reviewer 2 Report

If well the authors made quantitative analysis of the blots, the statistical analysis is still missing. How representative, variable and reproducible are those results? Statistical analysis is mandatory. The number of experiments carried out for every blot analysis need to be included too.

Author Response

If well the authors made quantitative analysis of the blots, the statistical analysis is still missing. How representative, variable and reproducible are those results? Statistical analysis is mandatory. The number of experiments carried out for every blot analysis need to be included too.

Answer: Thank you for pointing out. We've added the statistical analysis of all blots. All the western blots were repeated at least three times and highly reproducible.

Round 3

Reviewer 2 Report

I would like to thank the authors for the effort to include the statistical analysis.

I have no further comments or suggestions